# Humidity-tolerant porous polymer coating for passive daytime radiative cooling

Dongpyo Hong [1,7], Yong Joon Lee [2,7], Ok Sung Jeon[1], In-Sung Lee[1], Se Hun Lee [1], Jae Yeon Won[3,4], Young Pyo Jeon[1], Yunju La[1], Seonmyeong Kim[5,6], Gun-Sik Park[5], Young Joon Yoo [1] ✉ & Sang Yoon Park [1,3] ✉

Coating building envelopes with a passive daytime radiative cooling (PDRC) material has attracted enormous attention as an alternative cooling technique with minimal energy consumption and carbon footprint. Despite the exceptional performance and scalability of porous polymer coating (PPC), achieving consistent performance over a wide range of drying environments remains a major challenge for its commercialization as a radiative cooling paint. Herein, we demonstrate the humidity vulnerability of PPC during the drying process and propose a simple strategy to greatly mitigate the issue. Specifically, we find that the solar reflectance of the PPC rapidly decreases with increasing humidity from 30% RH, and the PPC completely losses its PDRC ability at 45% RH and even become a solar-heating material at higher humidity. However, by adding a small amount of polymer reinforcement to the PPC, it maintains its PDRC performance up to 60% RH, resulting in a 950% increase in estimated areal coverage compared to PPC in the United States. This study sheds light on a crucial consistency issue that has thus far been rarely addressed, and offers engineering guidance to handle this fundamental threat to the development of dependable PDRC paint for industrial applications.

Passive Daytime Radiative Cooling (PDRC), where Earth's heat is dissipated into space via sunlight reflection and blackbody radiation, has received considerable attention as an alternative cooling technology to mitigate global warming on the basis of its low energy consumption and environmental pollution[1–3]. Today, conventional vapor compression cooling technology is responsible for more than 10% of greenhouse gas emissions and at least 20% of power consumption, a vicious cycle that is accelerating global warming[4–7]. In response to this as a way to reduce the heat island effect and peak energy demand in densely populated cities, the integration of PDRC technology into the building envelope has been actively promoted by worldwide non-profit organizations such as the Cooling Roof Rating Council[8].

Motivated by the first demonstration of PDRC via a multilayered photonic structure in 2014[9], various studies to improve its scalability, reliability, and cooling efficiency in the form of films, fabrics, and structural materials have been reported[10–16]. Recently, PDRC paint has been gaining attention because it can be inexpensively applied to constructional objects such as buildings, towers, and pavements. For example, Li et al. reported that $BaSO_4$-acrylic paint exhibits excellent cooling performance with a daytime cooling temperature of 4.5 °C and an average cooling power of 117 W m$^{-2}$[17]. Wide bandgap pigments can prevent UV absorption and the low refractive index of such pigments can be overcome by increasing the filler ratio to nearly twice that of conventional white paint[4,18,19]. On the other hand, utilizing air pores

[1]Advanced Institute of Convergence Technology, Seoul National University, Suwon-si, Gyeonggi-do 16229, Republic of Korea. [2]PURITECH co., Ltd., Pyeongtaek-si, Gyeonggi-do 17745, Republic of Korea. [3]School of Electronic Engineering, Kyounggi University, Suwon-si, Gyeonggi-do 16227, Republic of Korea. [4]Department of Global Smart City, Sungkyunkwan University, Suwon-si, Gyeonggi-do 16419, Republic of Korea. [5]Department of Physics and Astronomy, Center for THz-Driven Biomedical Systems, Institute of Applied Physics, College of Natural Sciences, Seoul National University, Seoul 08826, Republic of Korea. [6]Mechatronics Research, Samsung Electronics Co., Ltd., Hwaseong-si, Gyeonggi-do 18448, Republic of Korea. [7]These authors contributed equally: Dongpyo Hong, Yong Joon Lee. ✉e-mail: youngjoonyoo@snu.ac.kr; yoonpark@kgu.ac.kr

instead of pigments as light scatterers has also been shown to provide excellent cooling performance, and has the advantages of not requiring a filler and providing flexible characteristics. The excellent cooling performance of PDRC paint results from efficient sunlight scattering by the hierarchical porous structure formed during the evaporation process of the porous polymer coating (PPC)[20]. More recently, studies on flexibility[21], durability[22], and coloration[23] toward the practical application of PPC-based PDRC paint have been conducted.

For the commercialization of PPC-based PDRC paint, the consistency of its cooling performance in various coating environments is expected to be a critical factor. Particularly in PPC, despite the aforementioned advantages, achieving consistent cooling performance is challenging because of the evaporation-sensitive nature of the PPC structure. The hierarchical porous structure of PPC formed by evaporation-induced phase separation (EIPS) is bound to be sensitive to the relevant evaporation conditions of the environment such as external humidity and temperature during the drying process. We found that the solar reflectance of PPCs reported to date has a very large deviation from 0.76 to 0.96, even among studies adopting a similar manufacturing method and coating thickness (Supplementary Fig. 1). These large deviation in the optical properties of PPCs are expected to result in an inconsistent cooling performance of PPC-based PDRC paint. Nonetheless, a detailed study on the cooling effect according to the evaporation environment has yet to be reported.

In this study, we developed a humidity-tolerant porous polymer based PDRC coating material by overcoming vulnerability to humidity in drying environments via mechanically reinforcing polymer matrix. While the conventional PPC drastically deteriorated when dried at above 30% RH and completely lost its daytime cooling abilities and even turn into the solar-heating material, fumed silica added PPC (FSPPC) maintained its cooling ability up to 60% RH humidity in the drying environment (cooling temperature of 7 °C at 30% RH, 3 °C at 60% RH). This improvement increased the areal coverage of the FSPPCs by 950% in the United States of America, from two to 43 applicable states. It is revealed that the superior consistency of FSPPC over that of PPC originates from its structural stability preventing the collapse of micropores during the drying process.

## Results
### Optical properties of PPC and FSPPC depending on drying-humidity

In order to realize daytime radiative cooling, high solar reflectance ($R_{solar}$) and thermal emittance in the atmospheric transmission window ($\varepsilon_{8-13}$) are essential to achieve absorbed solar energy less than the energy emitted as blackbody radiation. $R_{solar}$ and $\varepsilon_{8-13}$ are defined as

$$R_{solar} = \frac{\int_{0.28}^{2.5\mu m} I_{solar}(\lambda)R(\lambda)d\lambda}{\int_{0.28}^{2.5\mu m} I_{solar}(\lambda)d\lambda} \tag{1}$$

$$\varepsilon_{8-13} = \frac{\int_{8}^{13\mu m} I_{BB}(\lambda,T)\varepsilon(\lambda,T)d\lambda}{\int_{8}^{13\mu m} I_{BB}(\lambda,T)d\lambda} \tag{2}$$

where $\lambda$ is the wavelength, $I_{solar}(\lambda)$ is the ASTM G173-03 AM 1.5 solar spectrum, $R(\lambda)$ is the spectral reflectance of the sample, $I_{BB}(\lambda,T)$ is the spectral intensity for blackbody radiation at a temperature of $T$ calculated by Plank's law, and $\varepsilon(\lambda,T)$ is the spectral emittance of the sample. Poly(vinylidene fluoride-co-hexafluoropropene) [P(VdF-HFP)] has an almost negligible extinction coefficient in the solar spectral range and has multiple IR emitting vibrational modes at 8–13 μm, and thus is an ideal base material for radiative cooling[20]. In addition, the hierarchically porous structure of P(VdF-HFP)-based porous polymer coating (PPC) renders effective sunlight scattering, resulting in excellent solar reflectance. The pore formation in the

evaporation-induced phase separation (EIPS) process, however, can be greatly influenced by the external environment during the drying process[24–26]. As a result, a consistency issue in optical properties and cooling performance naturally arises as a practical paint should have spatiotemporal generality to respond to various conditions of the outdoor environment. In particular, it is important to understand the effect of humidity during the drying process on the structural and optical properties of the PPC considering that the atmospheric humidity varies spatiotemporally in an wide range, from almost 0 to 100% RH. To evaluate the PDRC performance consistency of PPC over such a wide range of humidity, optical properties of PPC dried at a range of humidity (hereafter referred to as drying-humidity) were investigated. In addition, we fabricated and characterized a fumed silica added porous polymer coating (FSPPC) that is aimed at improving the structural stability and performance consistency.

The reflectance over the solar spectral range (0.28–2.5 μm) for PPC and FSPPC dried at different humidity is shown in Fig. 1a, b. $R_{solar}$ at drying-humidity of 33% RH was found to be 95.7 ± 0.02%, similar to the highest range among the reported $R_{solar}$ of P(VdF-HFP) based PPC with a typical composition ratio among polymer, solvent, and nonsolvent and polymer (1:8:1) (Supplementary Fig. 1)[20,27,28]. However, as the drying-humidity increases, the reflectance drops in the entire solar spectral range, mainly led by a decline in the NIR region (Supplementary Fig. 2). The $R_{solar}$ of the PPC continuously decreases above approximately 30% RH with increasing drying-humidity, while it is almost constant at a lower range of humidity (Fig. 1c). At a drying-humidity of 67% RH, the $R_{solar}$ value of PPC decreases to just 77%, which is even much lower than that of commercial white paint. On the other hand, more interestingly, it is found that adding a small amount of fumed silica effectively alleviates the vulnerability to drying-humidity (denoted as fumed silica added PPC or FSPPC). Notably, in the case of FSPPC, the critical humidity is significantly elevated to ~60% RH, maintaining $R_{solar}$ above 94%. As will be further discussed in the later section, it was found that the solar reflectance was maximized at a fumed silica content of 5% mass fraction of the polymer, while excessive fumed silica degrades $R_{solar}$. Accordingly, all the experiments of this study were conducted with the content of fumed silica fixed at 5% mass fraction of the polymer.

The thermal emissivity, another major optical property that governs the cooling performance of PDRC, was investigated through IR spectroscopy (Fig. 2g). The thermal emissivity of PPC tended to increase as the drying-humidity increased, whereas the that of FSPPC did not show a significant change. In order to infer how the opposite changes in solar reflection and thermal emissivity according to dry humidity would comprehensively affect the PDRC performance, previously proposed figure of merit[19] for PDRC was estimated as follows (Fig. 1d).

$$\text{Figure of merit} = \varepsilon_{8-13} \cdot 10(1 - R_{solar}) \tag{3}$$

By estimating the figure of merit, we assessed the radiative cooling performance based on optical properties, independent of weather conditions or experimental setups. A larger positive value indicates higher cooling power under sunlight, while a larger negative value suggests higher heating power under sunlight. In the case of FSPPC, the figure of merit remained positive up to 60% RH, whereas for PPC, the figure of merit turned negative value when the drying-humidity exceeded 45% RH, indicating its transition into a solar-heating material under high humidity conditions. This result verifies the consistency issue regarding the vulnerability of PPC to drying-humidity and demonstrates that the addition of fumed silica can greatly alleviate this issue without sacrificing PDRC performance in the investigated humidity range.

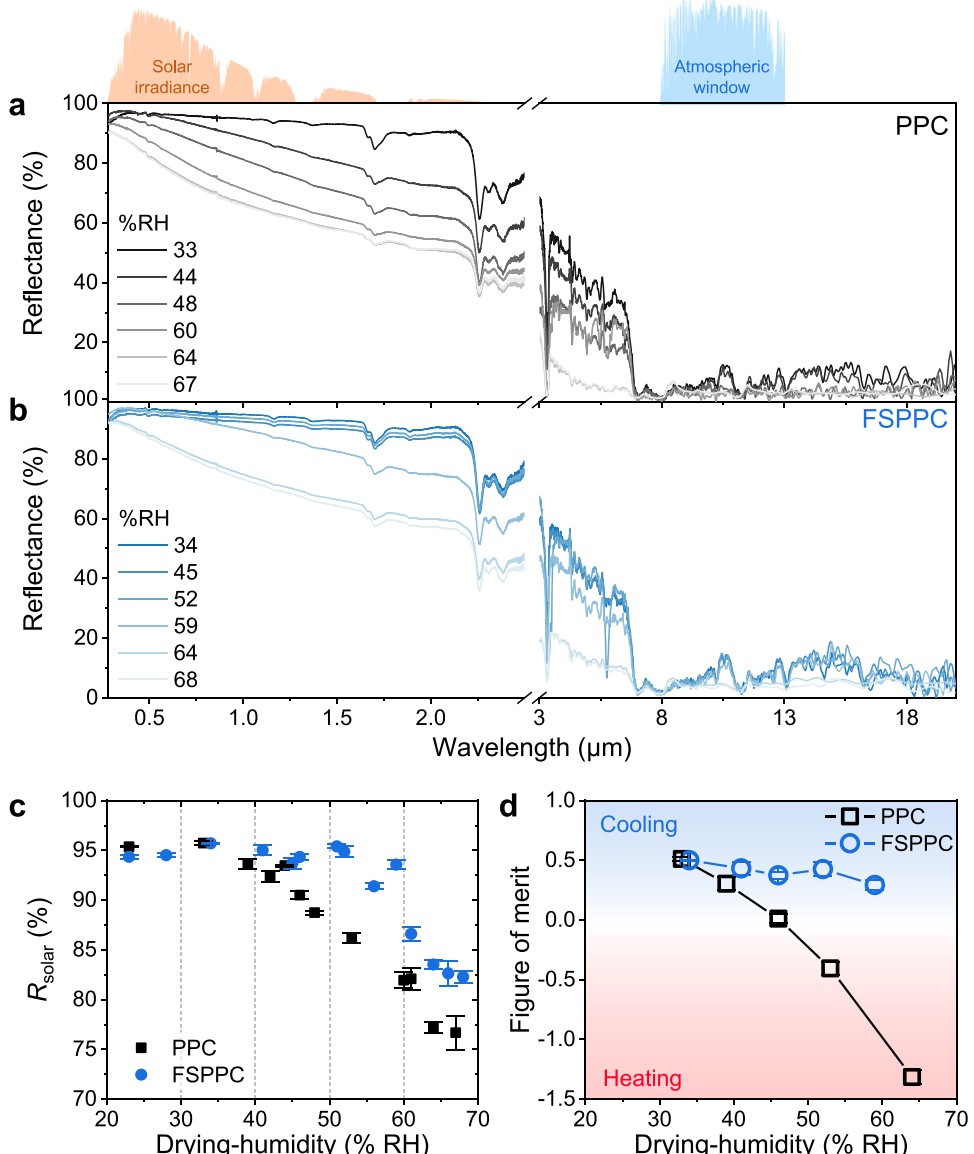

**Fig. 1 | Optical properties of porous polymer coating (PPC) and fumed silica added porous polymer coating (FSPPC) depending on the drying-humidity.** Measured spectral reflectance of PPC (**a**) and FSPPC (**b**) dried at various relative humidity. The normalized ASTM G173-03 solar spectrum and the transmittance at atmospheric window are depicted at the top of the graph. **c** Calculated solar reflectance ($R_{solar}$) weighted using the ASTM G173-03 solar spectrum. **d** Calculated figure of merit for PPC and FSPPC dried at different relative humidity. The error bars represent standard deviation of three independent measurements. Source data are provided as a Source Data file.

## Structural evolution depending on drying-humidity

To gain a deeper understanding of the sensitive nature of optical properties to the drying-humidity, we conducted a structural analysis for both PPC and FSPPC dried in the range of drying-humidity. Light scattering by porous structures highly depends on the geometry of individual pores[20,29]. Both of size and shape of the pore are involved in the scattering phenomenon and dramatically changes the radiative cooling performance[28]. We have focused on the micropore (diameter >0.5 μm) topology rather than the effect of nanopores in the wall of micropores (diameter <0.5 μm) due to the following reasons. First, the change in nanopores was not as noticeable as the change in micropores with regard to drying-humidity (Supplementary Fig. 3). Second, the change in solar reflectance was primarily observed in the visible to near-infrared region, where scattering by micropores play a dominant role. We further substantiated our findings through microscopic observation of the drying process of the casted PPC solution (Supplementary Fig. 4). As the solution undergoes phase separation during acetone evaporation, a polymer structure forms, succeeded by water evaporation, which leads to the emergence of surface micropores. The significant darkening of the image resulting from micropore formation verifies that their scattering has a prevailing impact on the overall film transmission, even within the visible light spectrum. Cross-sections of dried PPC and FSPPC were analyzed using a scanning electron microscope (SEM). Figure 2a, b show representative SEM cross-section images of the top microporous layer in PPC and FSPPC dried at different humidities. The diameter and aspect ratio of the pores change significantly as the drying-humidity increases. In the case of PPC, as the drying-humidity increases, pores become smaller and flatter. When dried at low humidity, the pore structures of PPC and FSPPC do not differ significantly, but as the drying-humidity increases, clear differences are seen. Unlike PPC, the shrinkage of pores with increasing humidity was relatively mitigated in the case of FSPPC. More specifically, although the dimension in the horizontal direction decreased in

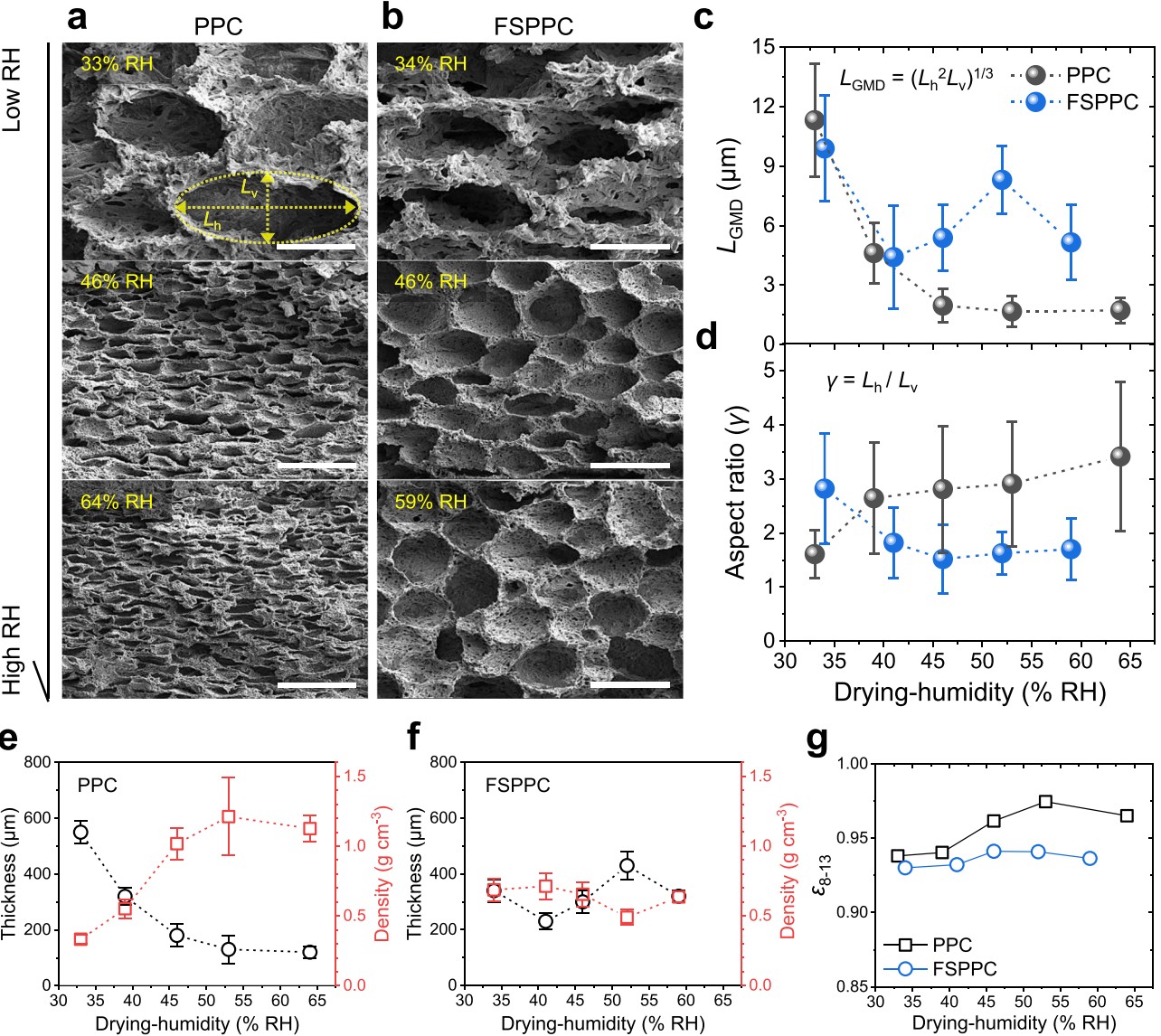

**Fig. 2 | Statistical analysis on the porous structure of porous polymer coating (PPC) and fumed silica added porous polymer coating (FSPPC).** Scanning electron microscope (SEM) cross-section images of (**a**) PPC dried at 33, 46, and 64 RH% and (**b**) FSPPC dried at 34, 46, and 59 RH% from top, respectively. Spheroidal modeling of the pore and definition of horizontal diameter ($L_h$) and vertical diameter ($L_v$) are illustrated in yellow ellipsoid. Structural parameters such as geometric mean diameter ($L_{GMD}$) (**c**) and aspect ratio $\gamma$ (**d**) of micropores in PPC and FSPPC depending on drying humidity. Error bar means the standard deviation of the size distribution. **e**, **f** Thickness and volumetric density of dried PPC and FSPPC depending on the drying-humidity. The error bars represent standard deviation of five independent measurements for each sample. **g** Thermal emissivity of PPC and FSPPC depending on the drying humidity. All scalebars in (**a**, **b**) are 10 μm. Source data are provided as a Source Data file.

some degree, that in the vertical direction remained comparatively constant.

As the first step to quantify the relationship between the porous structure and the optical properties, a statistical analysis of the micropore topology was conducted (Supplementary Fig. 5). Each pore is modeled as a spheroid with a horizontal diameter $L_h$ and a vertical diameter $L_v$. In the case of PPC, both $L_h$ and $L_v$ show a sharp decrease according to the drying-humidity. As the humidity increases from 33% RH to 64% RH, $L_v$ decreases tenfold from 8.5 μm to 0.8 μm while $L_h$ decreases from 13.2 μm to 2.6 μm. On the other hand, in the case of FSPPC, $L_h$ decreased less than that of PPC, and more interestingly $L_v$ showed an almost constant trend. As the drying-humidity increased from 34% RH to 59% RH, $L_v$ decreased slightly from 5.2 μm to 4.1 μm and $L_h$ decreased from 13.95 μm to 6.64 μm.

$L_h$ and $L_v$ can be converted to the geometric mean diameter ($L_{GMD}$)[30] and aspect ratio ($\gamma$) to analyze the effect of the pore size and shape on the optical performance distinctively. $L_{GMD}$ is defined as the diameter of a sphere having the same volume as the measured pore. $\gamma$ is defined as the ratio of $L_h$ to $L_v$ of the pore and was analyzed to determine the shape of the pore. The results of this analysis are depicted in Fig. 2c, d. The size of the micropores in both PPC and FSPPC decreased as the drying-humidity increases. In PPC, $L_{GMD}$ decreased to less than 2 μm at 45% RH, while in FSPPC, micropores did not shrink below a $L_{GMD}$ of 4.5 μm even at 60% RH. A clear difference in shape evolution was also observed between PPC and FSPPC. In the case of PPC, since $L_v$ decreased more steeply than $L_h$, $\gamma$ continuously increased from 1.6 to 3.4 as the relative humidity increases from 33% RH to 64% RH. Contrary to PPC, micropores in FSPPC shrank more in the horizontal direction than in the vertical direction, resulting in a

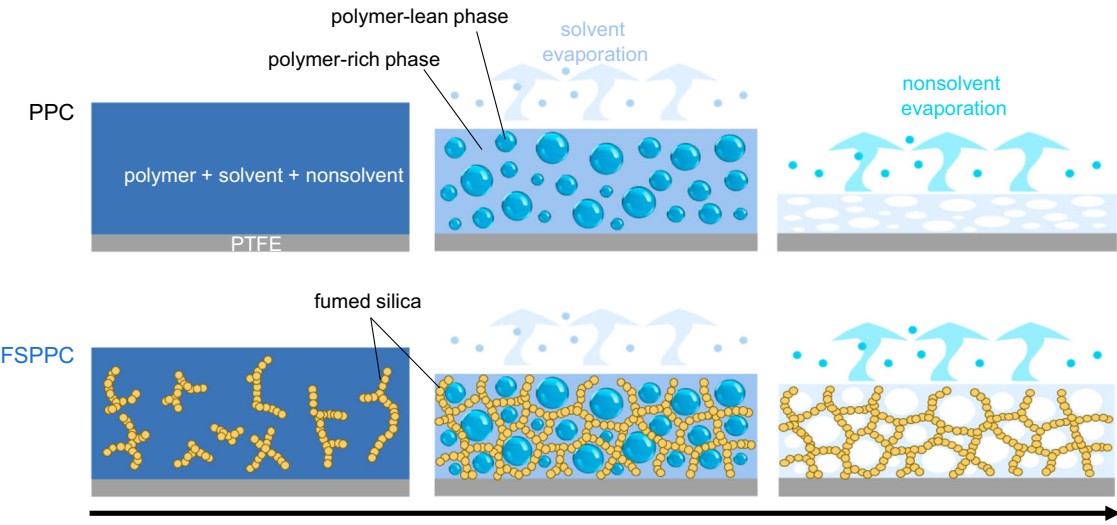

**Fig. 3 | Schematic illustration of pore formation in porous polymer coating (PPC) and fumed silica added porous polymer coating (FSPPC) during the drying process of the paint.** Initially, as the solvent evaporates, phase separation occurs, followed by the solidification of the polymer-rich phase, resulting in the formation of a porous polymer matrix. Subsequently, as the nonsolvent evaporates, pore formation takes place. In the case of PPC, pore collapse occurs during the drying process of the nonsolvent under high humidity conditions, while in the case of FSPPC, the embedded fumed silica prevents such collapse.

decreasing $\gamma$ with increasing drying-humidity. Hence, as the relative humidity increases, micropores in PPC became flatter ($\gamma$ greater than 1), whereas micropores in FSPPC remained spherical.

Changes in the micropore structure were accompanied by changes in the thickness and density of the entire film. With the fixed wet film thickness of 2 mm, the dry film thickness of PPC decreased from 550 μm to 120 μm as the drying-humidity increased, indicating a decrease in porosity of the films (Fig. 2e). In contrast, the thickness and density of FSPPC remained comparably constant up to 60% RH (Fig. 2f). A dense region (where only nanopores exist) was formed nearby the substrate probably due to the hydrophobicity of the substrate, but the thickness change of the dense region according to humidity was insignificant in both PPC and FSPPC. The decrease in thickness is mainly due to a decrease in the thickness of the microporous region. To sum up the structural analysis results, adding fumed silica greatly alleviates the volume decrease of individual pores and prevents the pore shape from flattening at high drying-humidity.

### Mechanism of structural consistency enhancement

The pore size reduction at high drying-humidity is thought to be due to fast phase separation into polymer-lean and polymer-rich phases, accompanied by structural quenching, originating from the slow evaporation rate of the non-solvent[31]. Rapid structural quenching may not provide enough time for coarsening of the polymer-lean phase domain, which becomes micropores in the final film. In addition, prolonged water evaporation after structural quenching may further deform the porous polymer network, while the addition of fumed silica mitigated such structural collapse under the drying process (Fig. 3).

Fumed silica has a fractal microstructure in which primary silicon dioxide particles several tens of nanometers in size form a secondary dendritic structure and possess high surface activity originating from the abundant hydroxyl groups (Fig. 4c, inset). High magnification scanning electron microscopy images of the FSPPC reveal that the fumed silica particles are evenly embedded in the inner walls of the micropores of the polymer matrix (Fig. 4b, c). The observation is in line with previous studies, which suggest that merely a few percent volume fraction of fumed silica is able to form a three-dimensional network within the polymer composite[32]. Fumed silica, which has been reported as an inorganic filler to mechanically reinforce polymer matrixes[33,34], may result in better structural stability of the structural scaffold

against gravitational and capillary forces with less structural deformation during the drying process. To verify the mechanical reinforcement of the polymer matrix by adding fumed silica, we conducted a compressive test on non-porous films with different fumed silica content (PC and FSPC) (Fig. 4d–f). The non-porous film was fabricated in the same manner as for PPC and FSPPC except that water was not added. As in the case of FSPPC, the fumed silica was evenly distributed throughout the FSPC (Fig. 4e). The compressive modulus and stress-strain-curve of the polymer composite matrix with different fumed silica content are shown in Fig. 5c, d. The compressive modulus of the FSPC was enhanced with fumed silica addition and maximized at content of 5 wt% of polymer. Notably, solar reflectance of FSPPC was also found to be maximized at content of 5 wt% of polymer, indicating that the aforementioned enhanced consistency of FSPPC originates from the mechanical reinforcement of the polymer composite matrix induced by embedding optimal content of fumed silica (Fig. 5a, b). Since silicon dioxide has a wide band gap of 8.9 eV, structural stability could be reinforced without sacrificing solar reflectance.

### Structure-optical property relation

To attain a deeper understanding of optical deterioration arising from structural collapse and to propose criteria for requisite structural stability, theoretical investigations into the structure-optical property relationship in such coatings have been conducted (see Supplementary Notes for more details). By incorporating the 3D scattering characteristics of individual pores obtained through numerical analysis into diffusion theory, the optical properties of the entire coating could be predicted with modest computational resources. In this approach, first, a FDTD simulation was carried out for a single isolated pore with various $L_{GMD}$ and $\gamma$, from which the scattering characteristics such as the scattering efficiency, scattering cross-section, scattering angular profile, and anisotropy factor could be extracted. Second, the obtained scattering characteristics are adopted into a diffusion model[35,36] to infer the transmittance through the slab in which micropores with specific $L_{GMD}$ and $\gamma$ are randomly distributed. Finally, the calculated transmittance as a function of $L_{GMD}$ and $\gamma$ is compared with the results of the above experimental analysis on optical properties of PPC and FSPPC according to average $L_{GMD}$ and $\gamma$ values.

The scattering efficiencies of a single isolated pore with varying $L_{GMD}$ and $\gamma$ averaged over the entire solar spectral range (0.28–2.5 μm)

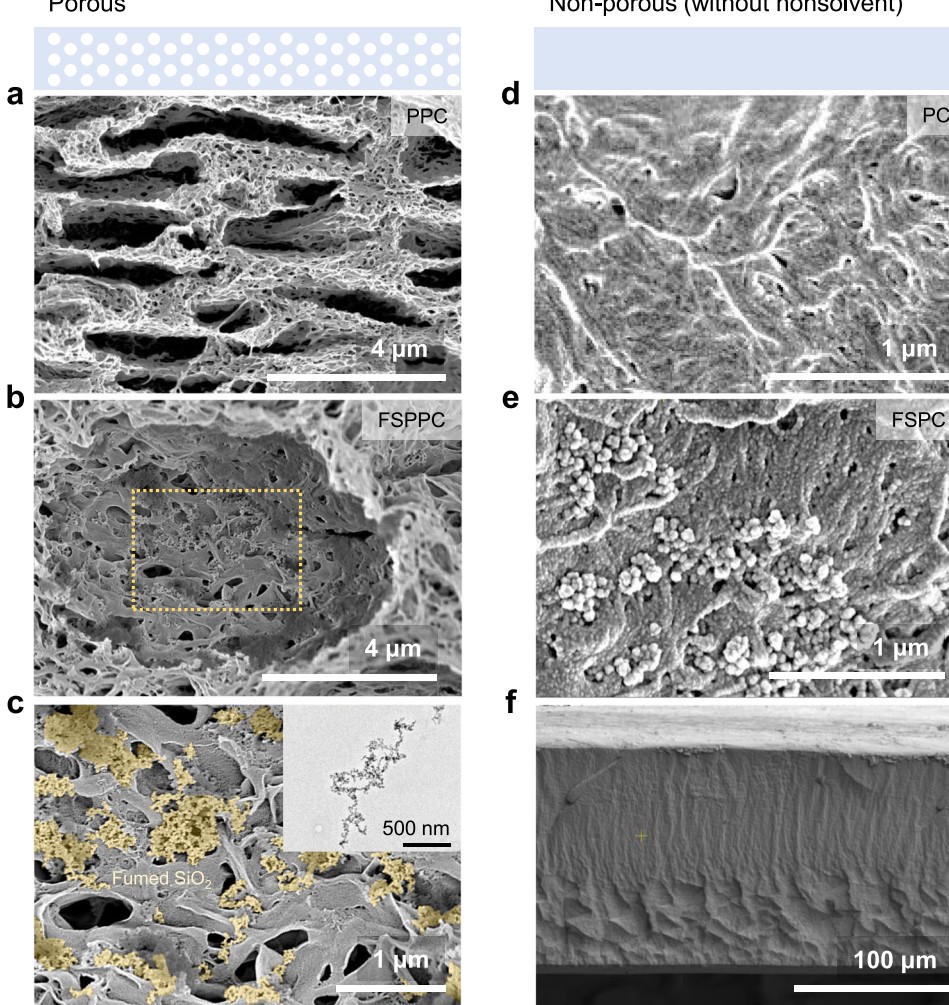

**Fig. 4 | The cross-section images of porous polymer film and non-porous polymer film. a, b** Cross-sectional scanning electron microscope (SEM) images of porous polymer coating (PPC) and fumed silica added porous polymer coating (FSPPC). **c** A high magnification SEM image showing well embedded fumed silica inside the pore wall. Inset shows the transmission electron microscope (TEM) image of individual fumed silica. **d, e** Cross-sectional SEM images of non-porous polymer matrix (PC) and fumed silica added non-porous polymer matrix (FSPC), which is a non-porous coating fabricated without the addition of a nonsolvent. **f** Low magnification cross-sectional SEM image of FSPC showing the non-porous, dense morphology through the entire film thickness.

are shown in Fig. 6a. The scattering efficiency drastically drops when $L_{GMD}$ of the pore is below 2 μm, regardless of $\gamma$. This decrease in the scattering efficiency is accompanied by the transition from the Mie scattering regime to the Rayleigh scattering regime as the pore size becomes smaller and approaches the wavelength of the NIR wavelength, as evidenced by the $\lambda^{-4}$ dependence of scattering efficiency and angular distribution (Fig. 6d and Supplementary Fig. 6)[37] Another important parameter, 1-<cosθ>, which indicates the direction of light deflection, averaged over the solar spectral range is shown in Fig. 6b. A value of 0 indicates perfect forward scattering, a value of 2 indicates perfect backward scattering, and a value of 1 indicates isotropic Rayleigh-like scattering. As $L_{GMD}$ decreases and $\gamma$ increases, this parameter tends to decrease. The decline in the value as $L_{GMD}$ decreases can be attributed to a tendency towards Rayleigh-like scattering as the wavelength becomes comparable or larger than the size of the pore. For $L_{GMD}$ larger than 2 μm, the spherical pore shows a larger 1-<cosθ> value than the oblate pore, due to an increase in the size of the lobe scattered at an angle of about 45 degrees, despite a decrease in backward scattering as the aspect ratio decreases. From the perspective of scattering directionality, it can be concluded that a small and spherical pore is the most advantageous.

To comprehensively predict the reflection characteristics with varying pore morphology, the transport mean free path in diffusion theory, which encompasses the scattering cross-section and directionality, was analyzed. The trend of scattering characteristic term $\sigma^{-1}(1-<cos\theta>)^{-1}$ within the transport mean free path with varying pore morphology was compared with that of measured optical characteristics (Fig. 6c). As $L_{GMD}$ decreases below 2 μm, the transport mean free path shows a rapid increase, resulting in deteriorated optical performance, while there is a relatively small difference depending on $\gamma$. In the case of PPC, $L_{GMD}$ falls below 2 μm as the drying-humidity increases, causing drastic increase in the transport mean free path. Meanwhile, in the case of FSPPC, $L_{GMD}$ does not drop below 4 μm even when dried at high humidity (~59% RH). Additionally, the strong agreement between the experiments and calculations indicates that changes in the scattering characteristics term is dominant over the changes in $\rho_s$ due to pore contraction. This is because the pore size and the overall film thickness decreases simultaneously with increasing humidity such that the number of pore layers in the vertical direction does not change substantially. Notably, when the number of pore layers throughout the film is inferred by dividing the thickness of the microporous region by the vertical diameter of an individual pore, it only exhibits a change

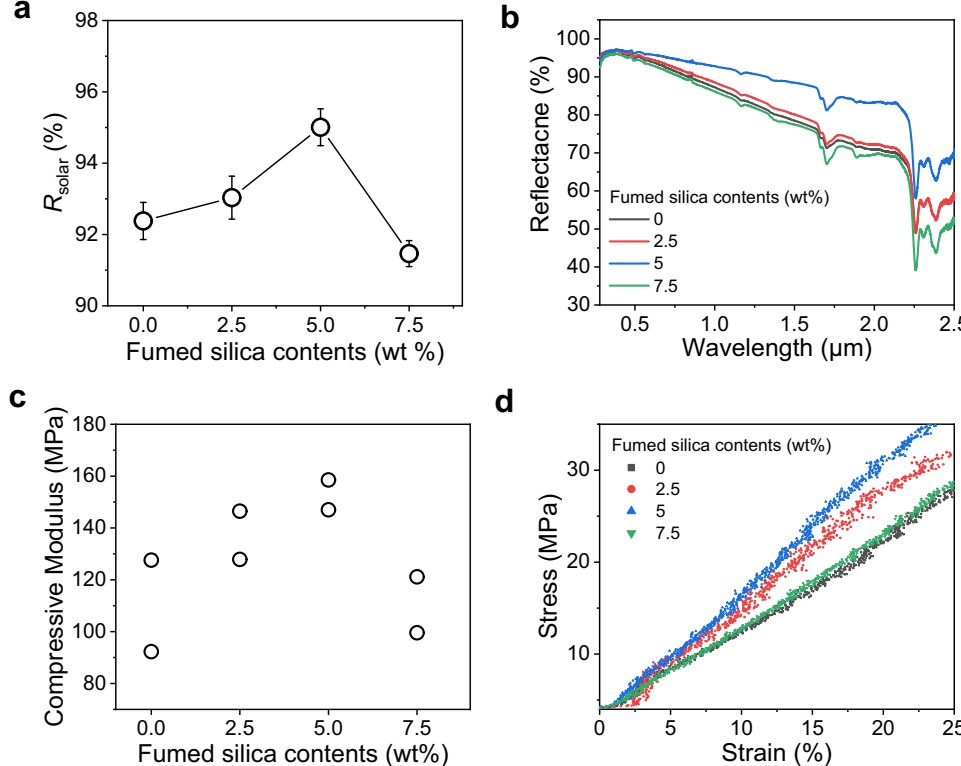

**Fig. 5 | The correlation between solar reflectance of the porous film and the mechanical strength of the polymer matrix.** The solar reflectance (**a**) and corresponding spectral reflectance (**b**) of porous polymer coating (PPC) and fumed silica added porous polymer coating (FSPPC) dried at a mid-drying-humidity (44–46% RH) depending on the fumed silica content. Compressive modulus measured by two independent experiments (**c**) and stress-strain curve (**d**) of non-porous polymer matrix (PC) and fumed silica added non-porous polymer matrix (FSPC) depending on the fumed silica content. The error bars represent standard deviation of three independent measurements for each sample. Source data are provided as a Source Data file.

within an order of number over the range of the drying-humidity (Supplementary Fig. 7), while the scattering cross-section and anisotropy factor show a few order changes. Therefore, the result reveals that it is crucial to retain the size of the micropores in the porous polymer to maintain consistent performance such that it does not fall below a critical size (~2 μm) during drying process.

## Outdoor cooling performance of the PPC and FSPPC

An outdoor test was conducted to investigate how the dependence of the structure-optical properties of PPC and FSPPC on humidity affects their radiative cooling performance (Fig. 7a, b). PPC and FSPPC were tested outdoors, and applied at low (~30% RH), moderate (~45% RH), and high (~60% RH) drying-humidities, and a summary of the results is presented in Fig. 7c, while corresponding cooling data are presented in Fig. 7d–f. Both FSPPC and PPC exhibited excellent daytime radiative cooling performance with cooling temperature at apex solar irradiance $\Delta T_{cool}$ of -7 °C when dried at low humidity. When dried at humidity above 45% RH, however, only FSPPC was capable of cooling during daytime, whereas PPC showed no cooling performance or even heated up above ambient temperature, which is consistent with optical figure of merit (Fig. 7c). Even when measured simultaneously for five consecutive days under various solar irradiation, this trend in cooling performance according to dry humidity was consistent (Supplementary Fig. 8). The industrial impact of the improvement of the humidity vulnerability is predicted based on the applicable area of PPC and FSPPC in the United States (Fig. 7g). PPC can be applied at annual average afternoon humidity of 30% or less, and FSPPC can be applied at 60% or less. According to our estimation, PPC is applicable only in two out of 50 dry states in the United States, while FSPPC is applicable in 43 states. The corresponding areal coverage of FSPPC is improved by 950% relative to that of PPC in the United States (from 610,150 km² to 6,396,558 km²). These findings highlight the importance of incorporating fumed silica into PPC to enhance its humidity stability and expand its applicable area, thereby contributing to its widespread industrial adoption and practical use with global impact.

## Discussion

In this study, the impact of drying-humidity on the optical properties and cooling performance of PPC was experimentally investigated. We demonstrate that the performance of conventional PPC rapidly deteriorates with increasing drying humidity. Based on an analysis of microporous structure and a theoretical analysis accompanied by a FDTD simulation, it was concluded that this deterioration was mainly due to the reduction in the size of macroscopic pores at high drying-humidity. Interestingly, our results suggest that the addition of fumed silica can greatly mitigate the vulnerability to drying-humidity. The addition of proper content of fumed silica reinforces the polymer matrix of the porous structure, which prevents structural collapse during the drying process. In the case of FSPPC, the figure of merit for radiative cooling of the dried film was found to be maintained up to 60% RH by preserving the size of the micropores such that it does not fall below the critical size. Accordingly, the applicable humidity range could be dramatically increased from 30% RH or less to 60% RH or less, and the applicable area range based on the United States could be increased by 950%. Given that the annual average humidity exceeds 50% RH in most countries, this study delineates performance consistency issues and provides crucial engineering guidance for the practical application and commercialization of PPC as a PDRC paint in real-world environments.

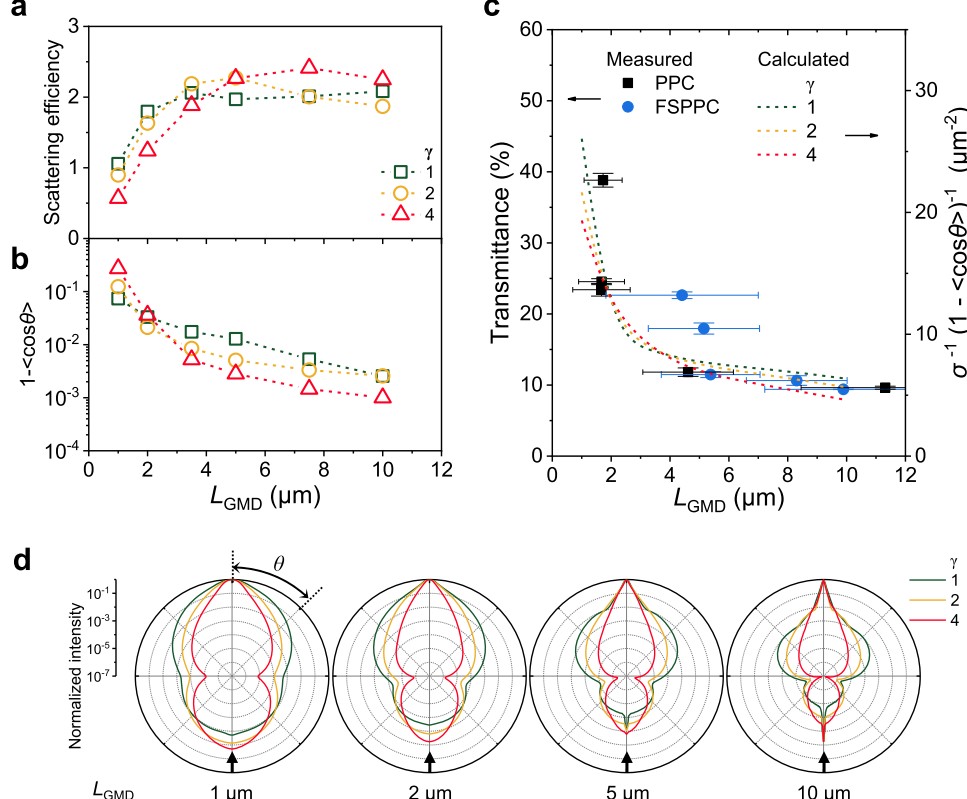

**Fig. 6 | Exploring the structure-property relation via a finite-difference time domain (FDTD) simulation. a** Anisotropy factor subtracted from one spheroid micropore in P(VdF-HFP) with different size and aspect ratio. **b** Simulated scattering efficiency of spheroid micropores in P(VdF-HFP) with different size and aspect ratio. **c** Comprehensive estimation of the trend of the transport mean free path compared with the experimental results. **d** Normalized scattering angular profile averaged over the whole solar spectrum range of spheroid pores in P(VdF-HFP) with different size and aspect ratio where the direction of wave incidence is denoted by a bold black arrow. The x- and y- error bars represent standard deviation of size distribution and that of three independent measurements for each sample, respectively. Source data are provided as a Source Data file.

## Methods

### Materials
Poly(vinylidene fluoride-co-hexafluoropropylene) ($M_w = 400{,}000$), fumed silica (99.5%, 10–20 nm(BET)) were purchased from Sigma-Aldrich. Acetone (EP grade, 99.5%) was purchased from DUKSAN PURE CHEMICALS Co., Ltd., South Korea.

### Fabrication of fumed silica added porous polymer coating (FSPPC) and porous polymer coating (PPC)
The fumed silica added P(VdF-HFP) porous polymer coating (FSPPC) solution was prepared through the following steps. Firstly, P(VdF-HFP) and fumed silica were dispersed in acetone at 40 °C. The mass fraction of fumed silica to P(VdF-HFP) ranged from 2.5 to 10%. Secondly, DI water was slowly added dropwise under vigorous stirring to form a transparent, homogeneous solution. The mass ratio of P(VdF-HFP)-acetone-water is 1:8:1. The prepared mixture was cast-dried in a PTFE mold with dimensions of $75 \times 75 \times 2$ (mm). For the PPC samples, the synthesis process was identical to that of FSPPC, except that fumed silica was not added. It is important to note that, considering practical application, we controlled the wet film thickness rather than the dry film thickness to prevent paint sagging and running. A wet film thickness of 2 mm was chosen based on the maximum range of wet film thickness notch gauges commercially available (ASTM D4414-95). Additionally, we conducted drying in an open state within a sufficiently large space equipped with a thermo-hygrostat to simulate real-world conditions where solvent evaporation does not influence atmospheric composition.

### Compressive test of non-porous polymer composite matrix
To uncover the origin of the structural stability in FSPPC, we investigated the compressive strength of the non-porous polymer composite matrix with varying fumed silica contents. The non-porous polymer matrix and the non-porous polymer-fumed silica composite matrix are denoted as PC and FSPC, respectively. The thickness of PC and FSPC was controlled to be around 300 μm, and they were shaped into discs with a 7 mm diameter. Subsequently, stress-strain curves were measured using a universal testing machine (TD-U01, TnDorf) equipped with compressive grips and a 200 kgf load cell. The compressive modulus of the discs was estimated from the slope of the stress-strain curve.

### Optical characterization
The optical properties of PPC and FSPPC were characterized in the solar spectral range (0.28–2.5 μm) and long-wavelength infrared spectral range (3–20 μm). For the visible to near-infrared spectral range (0.28–2.5 μm), the reflection of PPC and FSPPC films was measured using a UV-VIS-NIR spectrophotometer (LAMBDA 1050 + , PerkinElmer) with an integrating sphere. All data were calibrated against Spectralon standard samples provided by the manufacturer. For each sample, we conducted three reflection measurements, calculated the average value, and represented the standard deviation as an error bar. The spectral emittance of the films was calculated from the reflectance of the films measured within the wavelength of 3–20 μm using an FTIR spectrometer (INVENIO R, Brucker) equipped with a gold-coated integrating sphere.

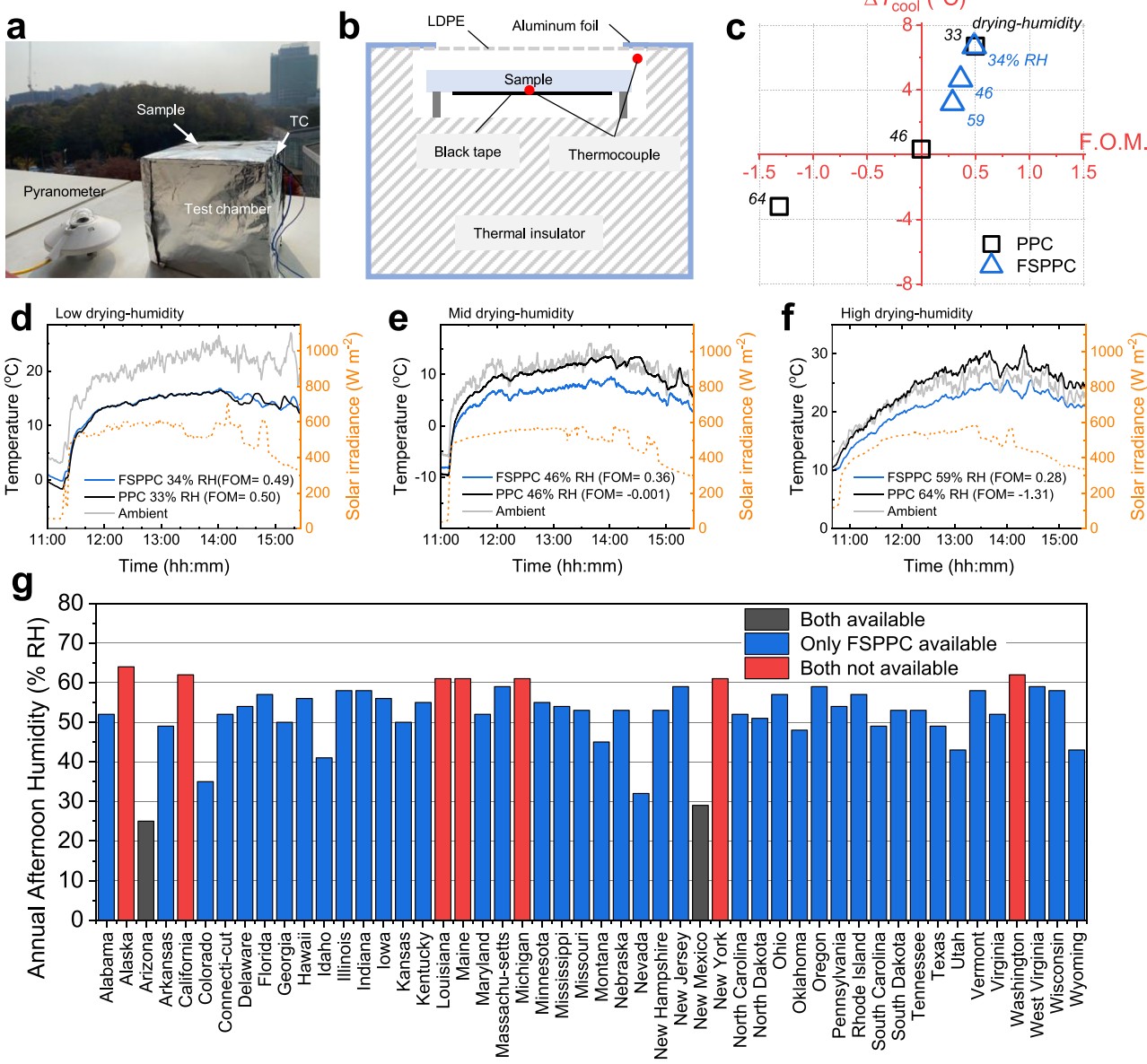

**Fig. 7 | All-day radiative cooling of fumed silica added porous polymer coating (FSPPC) coating dried at high humidity.** Optical photograph (**a**) and schematic (**b**) of the field test setup. The convection was shielded by a low-density polyethylene (LDPE) film. **c** Cooling performance and optical figure of merit (F.O.M.) of PPC and FSPPC according to the drying-humidity **d**–**f** Measured daytime radiative cooling of PPC and FSPPC dried at low humidity, moderate humidity, and high humidity, respectively. **g** Annual afternoon humidity of states in United States. Applicable areas of PPC and FSPPC are denoted by different colors. Source data are provided as a Source Data file.

## Structural characterization

Porous structure of PPC and FSPPC were investigated by scanning electron microscopy (SEM) using FEI Helios 5 UC instrument. To obtain clear cross-section images and minimize structural deformation resulting from fracturing, the films were fractured while immersed in liquid nitrogen. The geometry of more than 300 pores for each sample was statistically analyzed for calculating average and standard deviation of the distribution. The morphology of fumed silica was examined by transmission electron microscopy (TEM) using FEI Tecnai G2 F30 S-Twin instrument. The film thickness was measured using a digimatic micrometer (293-240-30, Mitutoyo). The weight of the 4 cm² square films was measured using a microbalance (MS204S, Mettler Toledo). Film density was calculated by dividing the weight by the volume, where the volume was determined by multiplying the thickness by the area.

## In-situ drying observation of PPC

PPC solution was casted onto a transparent mold with dimension of 20 × 20 × 1 (mm). As the solution dried, optical microscopy images were acquired using an OLYMPUS BX51TRF microscope with transmitted light mode. (Supplementary Fig. 4)

## Field test

The field tests were conducted using a test chamber (Fig. 6a, b) in Suwon, South Korea (37°17′39″N, 127°02′46″E, Elevation-107 m) on 18 December 2022 (ambient humidity-47% RH, clear sky, wind speed-3 m s⁻¹), 18 January 2023 (ambient humidity-41% RH, clear sky, wind speed-3.2 m s⁻¹), and 4 March 2023 (ambient humidity-41% RH, clear sky, wind speed-2.6 m s⁻¹) for Fig. 6d–f respectively. The above conditions referred to the weather at 14:00 on each day provided by the Korea Meteorological Administration. All measurements were

conducted under clear sky conditions. All Samples were tested in free-standing film form backed with black tape to absorb any transmitted light. A K-type thermocouple was sandwiched between black tape and sample to measure temperature of the sample. To minimize the effect of wind, convection was shielded by a low-density polyethylene (LDPE) film. Solar irradiance data were collected using a pyranometer (SMP11, Kipp&Zonen) located beside the test chamber.

## Data availability

The data that support the findings of this study are available from the corresponding authors upon request. Source data are provided with this paper.

## Code availability

The Lumerical simulation code is available from the corresponding author upon request.

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

## Acknowledgements

This research was supported by the National Research Foundation of Korea (NRF) grant funded by the Korea Government (MSIT) (Nos. 2016R1A3B1908336 to G.-S.P., 2020R1A2C2103137 to S.Y.P., and 2022R1C1C2011696 to S.H.L.), the Technology Innovation Program funded by the Ministry of Trade, Industry & Energy (MI, Korea) (No.

20020216-K_G012002021601-10054408 and RS-2023-00236794 to S.Y.P.), and the 「Gyeonggi-do Testbed Utilization Semiconductor Technology Development Project」 managed by the Advanced Institute of Convergence Technology with financial resources from Gyeonggi-do in year 2023 (AICT-06TB2). We thank E.J.Lee for her contribution in creating the illustration presented in Fig. 3.

## Author contributions
D.H., Y.J.Y., and S.Y.P. conceived the research and designed the experiments. D.H., Y.J.L., and J.Y.W. performed the experiments and analyzed the results. I.-S.L. performed the FDTD simulation. D.H. wrote the first draft of the manuscript. D.H., Y.J.L., Y.J.Y., S.Y.P., O.S.J., S.H.L., Y.P.J., S.K., and G.-S.P. discussed the study results and made substantial improvements to the original and revised manuscript.

## Competing interests
The authors declare no competing interests.
