## [Peer Review File · Nature Communications]

Humidity-tolerant porous polymer coating for passive daytime radiative coolingREVIEWER COMMENTS

Reviewer #1 (Remarks to the Author):

Very interesting article which can be considered for publication after minor revisions. See below for my comments:

1. Figure of merit = $\epsilon \tau - 10(1 - R_{\text{solar}})$, ϵ should not be subscript.

2. Figure 2(e) and (f). How is the density measured?

3. Lines 412-417. 'The field tests were conducted using a test chamber (Figure 5a-b) in Suwon, South Korea (37°17'39"N, 127°02'46"E, Elevation~107 m) on 18 December 2022 (ambient humidity~47% RH, clear sky, wind speed~3 m s⁻¹), 18 January 2023 (ambient humidity~41% RH, clear sky, wind speed~3.2 m s⁻¹), and 4 March 2023 (ambient humidity~41% RH, clear sky, wind speed~2.6 m s⁻¹) for figure 5a-c respectively.' The test time is winter or spring, when the temperature is lower. It is recommended to supplement the radiative cooling properties of samples in high temperature climates in summer.

Reviewer #2 (Remarks to the Author):

In this paper, the authors discuss the challenges related to consistent performance of porous polymer coating (PPC) across varying drying environments, emphasizing the detrimental impact of humidity on passive daytime radiative cooling (PDRC) ability. The study proposes a solution by incorporating a small amount of polymer reinforcement to maintain PDRC performance at higher humidity levels, significantly increasing estimated areal coverage. However, some points need clarifying and further justification. They are given as follows.

1. The authors chose porous polymer for PDRC coating. What are the prominent advantages of using PPC over other PDRC coating materials?

2. The authors mentioned the potential of PPC as a building envelope. In a scenario that PPC is applied to a building wall and its solvent is evaporated, the PPC will tend to flow downward until the solvent dries. In this case, what effect will it have on the size and shape of the pores of FSPPC? As a result, what effect will this have on optical characteristics and cooling performance?

3. The authors noted that the shape and size of the pores of PPC are related to the rate of nonsolvent evaporation. What about the porous structure formed under various wind speeds and ambient temperatures? Is FSPPC's fumed silica effective in addressing radiative cooling performance inconsistency problems caused by wind speed and temperature?

4. Typos should be corrected.

In the Methods section, on Page 22, line 413, "figure 5a-b" should be corrected to "figure 6a-b."
In the Methods section, on Page 22, line 473, "figure 5a-c" should be corrected to "figure 6d-f."

5. To accurately compare radiative cooling performance between samples with various drying humidity (for the same solar irradiance, wind speed, and humidity), field tests for three types of FSPPC should be conducted simultaneously at the same location.

RESPONSE TO REVIEWERS' COMMENTS

Response to Reviewer #1's Comments

Reviewer #1 (Remarks to the Author)

Very interesting article which can be considered for publication after minor revisions. See below for my comments:

Response: We thank for the positive assessment regarding the focus of our manuscript, and for all his/her useful suggestions.

1. Figure of merit = $\varepsilon_{8-13} - 10(1 - R_{\text{solar}})$, ε should not be subscript.

Response: We have made correction accordingly

Revision in the revised manuscript (Page 8, Line 131)

$$\text{Figure of merit} = \varepsilon_{8-13} - 10(1 - R_{\text{solar}}) \quad (3)$$

2. Figure 2(e) and (f). How is the density measured?

Response: First, the thickness of the 4 cm² square films prepared was measured at five different locations using a digimetic micrometer (Mitutoyo, 293-240-30, 1 μm resolution). The average thickness and standard deviation (error bars) were calculated based on these measurements. The average and deviation of the weight were determined by measuring three times on a microbalance (Mettler Toledo, MS204S, 0.1 mg resolution). The densities of the films were then calculated by dividing the weight by the volume, calculated by multiplying the thickness by the area of the film.

Additionally, for obtaining a more reliable standard deviation of the density, we recalculated the propagation of uncertainty as follows.

$$\rho_{\text{avg}} * \left(\Delta(\%)_{\text{volume}}^2 + \Delta(\%)_{\text{mass}}^2 \right)^{0.5}$$

The recalculated values are included in Figure 2e-f.

Figure 2. Statistical analysis on the porous structure of PPC and FSPPC. (a-b) SEM cross-section images of (a) PPC dried at 33, 46, and 64 RH% and (b) FSPPC dried at 34, 46, and 59 RH% from top, respectively. Spheroidal modelling of the pore and definition of horizontal diameter (L_h) and vertical diameter (L_v) are illustrated in yellow ellipsoid. (c-d) Structural parameters such as geometric mean diameter (L_{GMD}) (c) and aspect ratio γ (d) of micropores in PPC and FSPPC depending on drying humidity. Error bar means the standard deviation of the distribution. (e-f) Thickness and volumetric density of dried PPC and FSPPC depending on the drying-humidity. (g) Thermal emissivity of PPC and FSPPC depending on the drying humidity. All scalebars in (a-b) are 10 μm .

We also inserted the above measurement information into the Method part to enhance the clarity of the manuscript.

Revision in the revised manuscript (Page 22, Line 409-413)

Structural characterization

Porous structure of PPC and FSPPC were investigated by scanning electron microscopy (SEM) using FEI Helios 5 UC instrument. To obtain clear cross-section images and minimize structural deformation resulting from fracturing, the films were fractured while immersed in liquid nitrogen. The morphology of fumed silica was examined by transmission electron microscopy (TEM) using FEI Tecnai G2 F30 S-Twin instrument. **The film thickness was measured using a digimatic micrometer (293-240-30, Mitutoyo). The weight of the 4 cm² square films was measured using a microbalance (MS204S, Mettler Toledo). Film density was calculated by dividing the weight by the volume, where the volume was determined by multiplying the thickness by the area.**

3. Lines 412-417. ‘The field tests were conducted using a test chamber (Figure 5a-b) in Suwon, South Korea (37°17’39’’N, 127°02’46’’E, Elevation~107 m) on 18 December 2022 (ambient humidity~47% RH, clear sky, wind speed~3 m s⁻¹), 18 January 2023 (ambient humidity~41% RH, clear sky, wind speed~3.2 m s⁻¹), and 4 March 2023 (ambient humidity~41% RH, clear sky, wind speed~2.6 m s⁻¹) for figure 5a-c respectively. The test time is winter or spring, when the temperature is lower. It is recommended to supplement the radiative cooling properties of samples in high temperature climates in summer.

Response: Thanks for the meaningful suggestion. We strongly agree that the thermal stability of PDRC films is also crucial in practical applications. Unfortunately, the setup for the field test in this study began towards the end of the summer of 2022. Consequently, we were unable to acquire a substantial amount of data during the summer period. The following are preliminary data had obtained in the early stage of the field test setup, where there was a sunlight absorption problem of the chamber resulting in much higher chamber temperature than ambient. In this setup, we compared the cooling performance of Enhanced Specular Reflector Film (ESR, 3M), and FSPPC dried at 59% RH (Figure R1a). It was observed that the cooling performance of FSPPC was superior to ESR, primarily attributed to the difference in absorbance in the UV and NIR region between ESR and FSPPC (Figure R1b), implying the thermal stability of FSPPC.

Figure R1. (a) Field test conducted on 2022. 9. 8. (ambient humidity~ 50% RH, clear sky, wind speed~1.7 m s⁻¹) (b) Spectral reflectance of ESR and FSPPC (59% RH).

We are sorry for that we were unable to provide abundant summer data due to the timing of the research and submission. Instead, to enhance the reliability of our field test data, we have included the following additional field test data as follows, where FSPPC and PPC, dried at three different humidities each, were simultaneously measured consecutively for five days.

Revision in the revised supplementary materials (page 10)

Figure S8. Simultaneous field test of PPC(a) and FSPPC(b) dried at various humidity for consecutive five days.

Revision in the revised manuscript (Page 17-18, Line 334-336)

An outdoor test was conducted to investigate how the dependence of the structure-optical properties of PPC and FSPPC on humidity affects their radiative cooling performance (Figures 6a-b). PPC and FSPPC were tested outdoors, and applied at low (~30% RH), moderate (~45% RH), and high (~60% RH) drying-humidities, and a summary of the results is presented in Figure 5c, while corresponding cooling data are presented in Figures 6d-f. Both FSPPC and PPC exhibited excellent daytime radiative cooling performance with cooling temperature at apex solar irradiance ΔT_{cool} of ~7°C when dried at low humidity. When dried at humidity above 45% RH, however, only FSPPC was capable of cooling during daytime, whereas PPC showed no cooling performance or even heated up above ambient temperature, which is consistent with optical figure of merit (Figure 6c). **Even when measured simultaneously for five consecutive days under various solar irradiation, this trend in cooling performance according to dry humidity was consistent (Figure S8).** The industrial impact of the improvement of the humidity vulnerability is predicted based on the applicable area of PPC and FSPPC in the United States (Figures 6g-h). PPC can be applied at annual average afternoon humidity of 30% or less, and FSPPC can be applied at 60% or less. According to our estimation, PPC is applicable only in two out of 50 dry states in the United States, while FSPPC is applicable in 43 states. The corresponding areal coverage of FSPPC is improved by 1050% relative to that of PPC in the United States. These findings highlight the importance of incorporating fumed silica into PPC to enhance its humidity stability and expand its applicable area, thereby contributing to its widespread industrial adoption and practical use with global impact.

Response to Reviewer #2

In this paper, the authors discuss the challenges related to consistent performance of porous polymer coating (PPC) across varying drying environments, emphasizing the detrimental impact of humidity on passive daytime radiative cooling (PDRC) ability. The study proposes a solution by incorporating a small amount of polymer reinforcement to maintain PDRC performance at higher humidity levels, significantly increasing estimated areal coverage. However, some points need clarifying and further justification. They are given as follows.

Response: Thank you for carefully reviewing our research and providing an opportunity for in-depth and valuable discussions. We agree that consistency issues in PPC, particularly in relation to various environmental factors other than humidity, need to be addressed for practical applications of PPC. It is evident that further understanding of PPC dried at various environments is required in this regard. Based on our initial results, the application on the sidewall seems to require additional development, but interestingly, FSPPC has shown to enhance the stability of PPC not only against humidity but also in response to temperature and wind.

1. The authors chose porous polymer for PRDC coating. What are the prominent advantages of using PPC over other PDRC coating materials?

Response: Conventional TiO_2 pigments exhibit high scattering efficiency; however, the narrow bandgap of the pigments leads to strong UV absorption, limiting solar reflection of commercial TiO_2 -based white paint to below 85%. [Joule, **2020**, 4 (7), 1350.] To address this issue, a number of studies have explored the substitution of TiO_2 with wide bandgap pigments such as BaSO_4 , CaCO_3 . [ACS Applied Materials & Interfaces, **2021**, 13 (18), 21733., Cell Reports Physical Science, **2020**, 1 (10), 100221.] Nonetheless, there exists a trade-off relationship between refractive index and bandgap, with the refractive index of these pigments ranging from 1.5 to 1.7, similar to that of the polymer binder. Hence, simply embedding wide bandgap pigments into the polymer binder could not yield an enough solar reflectance for PDRC. As a result, researchers have increased the filler ratio above 60 vol%. Although the polymer-ceramic composite yields the high enough solar reflectance, this unconventional high

filler ratio not only raises costs of the paint but also imposes limits on the binder content, leading to a sacrifice of mechanical strength. As can be seen in the SEM images of these studies, the result is the formation of a lumpy, porous coating film to which particles are attached by a relatively small amount of binder. Indeed, reported cases of BaSO₄ paint have indicated lower abrasion resistance (wear index of 150) compared to commercial paint (wear index of 104). [ACS Applied Materials & Interfaces **2021**, 13 (18), 21733.] While this might pose more serious issues regarding flexibility and cracking, it is worth noting that PPC composed entirely of polymers, exhibits excellent flexibility. To visualize this, we also attach an origami photo of a typical PPC.

Figure R2. Origami photo of the typical free-standing PPC films.

2. The authors mentioned the potential of PPC as a building envelope. In a scenario that PPC is applied to a building wall and its solvent is evaporated, the PPC will tend to flow downward until the solvent dries. In this case, what effect will it have on the size and shape of the pores of FSPPC? As a result, what effect will this have on optical characteristics and cooling performance?

Response: Thank you for the insightful comments. We strongly agree that research on the applications of radiative cooling coating to building sidewalls is highly important and should be pursued. We've further investigated the influence of gravity and explored the applicability to the sidewall of a building by casting and drying solutions on inclined substrates. We measured the film thickness, porous structure, and solar reflectance of the films. The drying behavior of PPC and FSPPC on the inclined substrate was examined from two perspectives.

Firstly, there was a decrease in wet film thickness due to limited viscosity and a substantial decrease in film thickness. Secondly, changes in pore structure during the drying process due to gravitational force and downward flow were observed.

Both PPC and FSPPC exhibited a tendency for the film thickness to decrease with the inclination of the substrate. In the case of PPC, even with a substrate inclination of only 30° , the thickness decreases more than ten-fold to be $\sim 35 \mu\text{m}$. For FSPPC, due to the relatively high viscosity, it was much thicker ($\sim 100 \mu\text{m}$) than PPC at same angle of inclination, probably because of slightly higher viscosity of FSPPC than PPC. However, for the vertical substrate, both PPC and FSPPC exhibit a decrease in thickness to less than $20 \mu\text{m}$. It is important to note that, since formation of both PPC and FSPPC utilize phase inversion, there is a limit to increasing the ratio of polymer to solvent in the liquid phase to prevent solidification before casting, resulting in a low viscosity of casting solution. This limited polymer contents not only reduces the wet film thickness but also limits dry film thickness to wet film thickness ratio, resulting in substantial thinning of the films on inclined surfaces.

Figure R3a shows spectral reflectance of PPC and FSPPC dried on inclined substrates with angle θ . As expected, the solar reflectance of PPC was found to decrease to 69% and 50% as the inclination of the substrate increased to 30° and 90° . Meanwhile, the solar reflectance of FSPPC was found to decrease to 84% and 52% as the inclination of the substrate increased to 30° and 90° , suggesting that further engineering is inevitable for radiative cooling coating on building sidewalls for both of PPC and FSPPC.

As the substrate tilts, the initial phase separation is thought to occur while the solution is flowing, and there will be a change in the direction of gravitational force of $g \cdot \sin\theta$ parallel to the substrate during the process of non-solvent removal after phase separation. We observed cross-sections of the films using SEM to investigate the effects of flow and gravity on pore size and shape (Figure R3b). Both FSPPC and PPC showed a tendency for the shape of the pores to become irregular and flattened when compared to drying at $\theta = 0$. An interesting point is that even when the substrate is tilted, the direction of flattening is perpendicular to the substrate, implying that the gravitational force is not dominant cause of the observed pore collapse.

Figure R3. Effect of tilting substrate. Reflectance spectra (a) and SEM cross-section image (b) of PPC and FSPPC coated on substrate with various angle between substrate and horizontal.

To explore the feasibility of applying porous polymer coating with phase inversion to the sidewall, additional experiments were conducted by partially substituting polymers to increase the viscosity of the solution. Figure R4 shows the preliminary results. Ethyl cellulose (EC), a widely used thickening agent, was used to substitute 50% of PVDF-HFP, leading to a

significant increase in viscosity of the paint. As a result, the thickness of EC added PPC (ECPPC) could be comparably maintained even when dried at vertical substrate ($\theta = 90^\circ$), and it exhibited a significantly increased reflectance compared to PPC and FSPPC, confirming that reflectance enhancement through thickening is possible ($R_{\text{solar}} = 76\%$). This was thought to be attributed to much larger film thickness with maintaining porous structure. However, UV absorption of EC greatly limit the solar reflectance of ECPPC, and further research is planned to explore more suitable thickeners and compositions to achieve highly reflective sidewall PPC coating enough for daytime radiative cooling.

Figure R4. Possible engineering of PPC for its sidewall application. a) Spectral reflectance of ECPPC showing significantly higher reflection than PPC and FSPPC over all solar spectral range. b) SEM cross-section image of ECPPC coated on a vertical substrate. Low-magnification image of the whole film (top) and topmost pore layers of ECPPC (bottom).

Additionally, we need to note that, when applying radiative cooling paint to building sidewalls, considerations should extend beyond the film's thickness and reflectance. The impact of terrestrial infrared radiation, also known as earth glow, emanating from the earth surface must be considered, which has gained significant attention recently. [arXiv:2006.11931, last revised 2021] Achieving effective cooling performance on sidewalls necessitates additional properties, such as selectivity or directionality of IR emission. Research in this aspect within the field of radiative cooling coatings is still relatively limited. Implementing such properties (selectivity, directionality) in large area, especially in randomly distributed systems like PPC, is thought to be a highly challenging task.

3. The authors noted that the shape and size of the pores of PPC are related to the rate of nonsolvent evaporation. What about the porous structure formed under various wind speeds and ambient temperatures? Is FSPPC's fumed silica effective in addressing radiative cooling performance inconsistency problems caused by wind speed and temperature?

Response: To investigate the influence of temperature, we dried PPC and FSPPC in a constant temperature and humidity chamber (CKFT-800SS, CKSI, Korea) and subsequently examined their optical performance and structure (Figure R5). As depicted in Figure R5a, the spectral reflectance of PPC and FSPPC deteriorated at elevated temperatures, particularly in the near-infrared (NIR) region. Notably, for FSPPC, there was no significant decrease after 40°C, whereas for PPC, there was a sharp decline in reflectance as the temperature increased to 60°C. Concurrently, a decrease in film thickness was observed with increasing drying temperature, indicating a reduction in porosity. Like observations under high drying-humidity conditions, high drying temperature induced the collapse of the porous structure (Figure R5b, inset). Indeed, the SEM cross-section image (Figure R5c-d) reveals a drastic collapse of porous structures at high drying temperatures. While FSPPC also experienced pore collapse, it could maintain larger pore sizes than PPC, potentially contributing to the observed differences in reflectance at 60°C. These observations suggest that the addition of fumed silica serves to enhance the structural-optical stability of PPC, not only against variations in drying humidity but also under different drying temperatures.

Figure R5. Effect of drying-temperature. Reflectance spectra (a), Solar reflectance (inset: thickness) (b) and SEM cross-section images of PPC (c) and FSPPC (d) dried at various environment temperature.

During the drying process of PPC, wind blowing on the surface can accelerate the evaporation of solvents and nonsolvent. This not only alters the compositional path during the phase inversion process (especially at nearby the surface) but also induces convection flow within the solution, potentially impacting the overall pore structure. Therefore, we observed the structure and optical properties of PPC dried under various wind speeds by adjusting the power of the circulator and the circulator-substrate distance (Figure R6a). Like other environmental factors, wind also acts as a detrimental factor reducing the solar reflectance of both PPC and FSPPC (Figure R6b). In the case of PPC, the solar reflectance dropped from the original 95%

to 91% when the wind speed reached 4 m/s. With increasing wind speed, it was observed that the microporous regions of PPC gradually narrowed (Figure R6c) and PPC became more asymmetric. The individual pores in the upper layer also significantly reduced in size, primarily influencing the decrease in reflectance in the NIR region. For FSPPC, although the pore sizes were similar to PPC, the microporous regions were relatively wider. Consequently, FSPPC exhibited a comparatively smaller decrease in reflectance ($R_{\text{solar}} \sim 94\%$ at 4 m/s of wind speed). An interesting observation was the formation of a single layer of relatively larger pores in the topmost layer at a wind speed of 4 m/s. This might contribute to FSPPC showing higher NIR reflectance at 4 m/s compared to 2 m/s. Although the underlying mechanisms are not fully elucidated here, it is believed to be closely related to convection occurring and the rapid solidification nearby the surface. Further in-depth research on this interesting aspect is planned for subsequent studies.

Figure R6. Effect of wind speed. (a) Schematic and photograph of experimental setup for examining wind effect on drying of PPC and FSPPC. (b) Spectral reflectance, (c) SEM cross-section images of the fully dried films.

4. Typos should be corrected.

In the Methods section, on Page 22, line 413, "figure 5a-b" should be corrected to "figure 6a-b."

In the Methods section, on Page 22, line 473, "figure 5a-c" should be corrected to "figure 6d-f."

Response: We have made correction accordingly

Revision in the revised manuscript (Page 22, Line 419, 423)

Field test

The field tests were conducted using a test chamber (**Figure 6a-b**) in Suwon, South Korea (37°17'39''N, 127°02'46''E, Elevation~107 m) on 18 December 2022 (ambient humidity~47 %RH, clear sky, wind speed~3 m s⁻¹), 18 January 2023 (ambient humidity~41% RH, clear sky, wind speed~3.2 m s⁻¹), and 4 March 2023 (ambient humidity~41% RH, clear sky, wind speed~2.6 m s⁻¹) for **figure 6d-f** respectively. The above conditions referred to the weather at 14:00 on each day provided by the Korea Meteorological Administration. All measurements were conducted under clear sky conditions. All Samples were tested in free-standing film form backed with black tape to absorb any transmitted light. A K-type thermocouple was sandwiched between black tape and sample to measure temperature of the sample. To minimize the effect of wind, convection was shielded by an LDPE film. Solar irradiance data were collected using a pyranometer (SMP11, Kipp&Zonen) located beside the test chamber.

5. To accurately compare radiative cooling performance between samples with various drying humidity (for the same solar irradiance, wind speed, and humidity), field tests for three types of FSPPC should be conducted simultaneously at the same location.

Response: Thanks for your suggestion to enriching the content of the manuscript. We expanded the test chamber and simultaneously measured PPC and FSPPC samples dried at different humidity (used in Figure 6d-f) for a duration of 5 days each. This allowed us to reaffirm the trends observed in Figure 6. Even with simultaneous measurements, for PPC, it was observed that when dried at 34% relative humidity (RH), it exhibits cooling performance, but at 46% RH, the cooling performance is lost, and at 64% RH, a tendency for radiative heating is maintained. This is primarily attributed to the decrease in solar reflectance of PPC with drying humidity. On days with low solar irradiance due to cloudy weather, this deviation becomes less pronounced, implying the deviation in their cooling temperature were originated from the deviation in their solar reflectance. On the other hand, for FSPPC, cooling performance was maintained at 33% RH, 46% RH, and 59% RH.

Revision in the revised supplementary materials (Page 10)

Figure S8. Simultaneous field test of PPC(a) and FSPPC(b) dried at various humidity for consecutive five days.

Revision in the revised manuscript (Page 17-18, Line 334-336)

An outdoor test was conducted to investigate how the dependence of the structure-optical properties of PPC and FSPPC on humidity affects their radiative cooling performance (Figures 6a-b). PPC and FSPPC were tested outdoors, and applied at low (~30% RH), moderate (~45% RH), and high (~60% RH) drying-humidities, and a summary of the results is presented in Figure 5c, while corresponding cooling data are presented in Figures 6d-f. Both FSPPC and PPC exhibited excellent daytime radiative cooling performance with cooling temperature at apex solar irradiance ΔT_{cool} of ~7°C when dried at low humidity. When dried at humidity above 45% RH, however, only FSPPC was capable of cooling during daytime, whereas PPC showed no cooling performance or even heated up above ambient temperature, which is consistent with optical figure of merit (Figure 6c). **Even when measured simultaneously for five consecutive days under various solar irradiation, this trend in cooling performance according to dry humidity was consistent (Figure S8).** The industrial impact of the improvement of the humidity vulnerability is predicted based on the applicable area of PPC and FSPPC in the United States (Figures 6g-h). PPC can be applied at annual average afternoon humidity of 30% or less, and FSPPC can be applied at 60% or less. According to our estimation, PPC is applicable only in two out of 50 dry states in the United States, while FSPPC is applicable in 43 states. The corresponding areal coverage of FSPPC is improved by 1050% relative to that of PPC in the United States. These findings highlight the importance of incorporating fumed silica into PPC to enhance its humidity stability and expand its applicable area, thereby contributing to its widespread industrial adoption and practical use with global impact.

REVIEWERS' COMMENTS

Reviewer #1 (Remarks to the Author):

The revised manuscript could be accepted now.

Reviewer #2 (Remarks to the Author):

The authors have addressed all my comments. I thus recommend its publication.